# Using Noninvasive Electrophysiology to Determine Time Windows of Neuroprotection in Optic Neuropathies

**DOI:** 10.3390/ijms23105751

**Published:** 2022-05-20

**Authors:** Vittorio Porciatti, Tsung-Han Chou

**Affiliations:** Bascom Palmer Eye Institute, University of Miami, Miami, FL 33136, USA; tchou@med.miami.edu

**Keywords:** neuroprotection, retinal ganglion cells, optic neuropathy, glaucoma, pattern electroretinogram

## Abstract

The goal of neuroprotection in optic neuropathies is to prevent loss of retinal ganglion cells (RGCs) and spare their function. The ideal time window for initiating neuroprotective treatments should be the preclinical period at which RGCs start losing their functional integrity before dying. Noninvasive electrophysiological tests such as the Pattern Electroretinogram (PERG) can assess the ability of RGCs to generate electrical signals under a protracted degenerative process in both clinical conditions and experimental models, which may have both diagnostic and prognostic values and provide the rationale for early treatment. The PERG can be used to longitudinally monitor the acute and chronic effects of neuroprotective treatments. User-friendly versions of the PERG technology are now commercially available for both clinical and experimental use.

## 1. Introduction

The death of retinal ganglion cells (RGCs) and their axons is the final common pathway of optic neuropathies resulting in loss of vision [1,2]. Neuroprotective strategies aimed at preventing loss of RGCs and sparing their function have been an area of intense investigation in animal models [3,4,5]. The great majority of experimental studies on neuroprotective strategies have been performed in glaucoma models using a large variety of neuroprotectants targeting multiple molecular pathways, often with impressive positive effects [2,6]. While neuroprotection studies in experimental models provide powerful proofs of principle, translation of neuroprotective strategies to the clinical application remains elusive [7,8,9]. One caveat of experimental models is that they are a gross approximation of the corresponding clinical condition [10,11], resulting in limited concordance of treatment effects between preclinical models and clinical trials. Another limitation is that results obtained in animal models most often reflect neuroprotective protocols started in temporal proximity of the induction of the pathological condition, while in the clinical condition therapeutical options are generally initiated after diagnosis that may occur relatively late over the course of the disease. A further limitation is that the sophisticated methods to assess RGC structure and function in experimental models are not generally applicable in the clinical setting. Here, we offer a perspective on the optimal time window for neuroprotective treatments to rescue RGC from death and preserve their function based on noninvasive methods to assess RGC functional integrity that can be used both in experimental models and clinical trials.

## 2. The Tipping Point

In progressive optic neuropathies, the tipping point represents the idealized transition from a physiological state to a pathological state. During the period preceding the tipping point (critical period) [12,13,14] accumulating adverse factors eventually overwhelm homeostatic mechanisms and cause irreversible and progressive cell death. The duration of the critical period of transition can be of the order of years, as in glaucoma, [15] or months, as in Leber’s Hereditary Optic Neuropathy (LHON) [16], and its identification would provide a red flag of impending disease and an opportunity to consider neuroprotective treatment in a time window where altered conditions may be still capable of reversal. While the tipping point is a well-established intuitive concept (Figure 1), its identification is challenging as phenotypic expression and molecular changes occurring during the critical period overlap with those of the normal condition, and homeostatic neuroplasticity mechanisms to maintain normal vision offset pathological alteration [17]. Later stages are dominated by cell survival and associated maladaptive processes including rewiring of the neural tissue and disruption of function that define the manifest disease state [18]. As sketched in Figure 1, there are several potential therapeutical time windows for neuroprotection, each of them probably resulting in a different outcome. Prophylactic neuroprotection (Rx_t0 in Figure 1) based on risk factors only is not currently considered in the clinical setting [19]. Typically, neuroprotective actions are considered by the time the disease is manifest (Rx_t3, Rx_t4 in Figure 1) [20,21] with the goal of slowing further damage. If a goal of neuroprotective therapy is to preserve RGC integrity and have a long-term efficacy, then it should be initiated as early as possible, ideally at pre-clinical stages (Rx_t1, Rx_t2, in Figure 1) where adaptive neural mechanisms may be still reversible. At preclinical stages, noninvasive structural RGC and RNFL assessments are unlikely to provide meaningful clinical indications [22,23]. In contrast, adaptive changes occurring during the critical period may impair the electrophysiological response of RGCs to visual stimuli, which can be used as biomarker of impending disease, to monitor its progression, and to provide a rationale for initiating neuroprotective treatment.

## 3. Electrophysiological Testing of Retinal Ganglion Cell Function

The electrophysiological activity of RGCs and their axons can be tested with specific variants of the electroretinogram (ERG) [24]. The best-understood and most sensitive technique is the ERG in response to contrast-reversing patterns (Pattern Electroretinogram, PERG). While the precise cellular sources of the PERG signal are not known, the PERG depends on the presence of functional RGCs, as it is rapidly abolished after the optic nerve crush that results in RGC degeneration, while the standard ERG remains unaffected. Both spiking and nonspiking electrical activity contribute to the PERG. Compared to the standard ERG, the PERG has a much smaller amplitude. However, using state-of-the-art equipment with robust averaging and processing to improve the signal-to-noise ratio, the PERG can now be easily recorded from surface adhesive electrodes in human and subdermal electrodes in mice (Figure 2) [25,26].

The PERG may be altered before histological loss of RGCs in glaucoma and optic neuropathies in both human and animal models [27,28,29]. The PERG can also inform about the response dynamics over a range of visual stimuli of different strength [30] as well as the ability to autoregulate under physiologically stressful conditions such as body inversion [31,32] or flicker-induced increase in metabolic demand [33]. Both response dynamics and autoregulation may provide useful biomarkers to establish altered RGC function not associated with cell death [30,34]. Typically, neuroprotection studies in experimental models of optic neuropathies quantify the effect of treatment by comparing the RGC/axon number of an independent control group with that of the study group at a given endpoint. Noninvasive electrophysiology such as PERG provides longitudinal information on overall RGC function from baseline to endpoint, and additionally it provides unique information on the acute effect of treatment and the time course of the effect, which includes the potential neuroenhancement effect as well as the potential toxic effect and is useful for screening purposes.

## 4. Comparing RGC Function with RGC Number

A strong proof of concept for the use of PERG as a biomarker of premanifest disease is offered by the DBA/2J mouse strain, which spontaneously develops a pigment-liberating iris disease, resulting in age-related IOP elevation and glaucoma [35,36]. Figure 3A compares the time course of IOP, PERG amplitude, and optic nerve axon number as a function of age of DBA/2J mice [29]. The IOP increases moderately between 2 and 7 months, and more sharply thereafter, when the optic nerve starts losing axons. By the time axon loss is noticeable at about 8 months of age, the PERG signal has already lost over 50% of baseline amplitude at 2 months of age. This indicates that RGCs become dysfunctional before they die. Multiple regression analysis of data shown in Figure 3A reveals that age (Log p = 28.5) plays a larger role than IOP (Log p = 3.8) in progressive loss of PERG signal. The horizontal distance between the decay curves of function and structure provides an estimate of the lifespan of sick RGCs, which represents the time window of opportunity for treatment to prevent RGC death. The vertical distance between the decay curves of function and structure provides an estimate of RGC dysfunction that is not accounted for by cell death, which is potentially reversible [37].

The comparison between the time courses of PERG amplitude and axon number (Figure 3A) offers an opportunity to investigate the relationship between RGC dysfunction and death [13]. The working hypothesis assumes that at any given time point the residual PERG amplitude reflects the summed contribution of still normal RGCs, the reduced contribution of sick RGCs, and the null contribution of dead/lost RGCs, each in relative proportions. The residual axon count reflects the remaining number of RGCs. The hypothesis also assumes that at each successive timepoint a constant proportion of RGCs becomes sick (decay rate **b**), functions at reduced capacity (dysfunction coefficient **d**) and survives for a limited amount of time (time lag **τ** between sick and dead RGCs). These events will be reflected in progressive loss of PERG amplitude and axon number, with the former expected to anticipate loss compared to the latter. These parameters can be included in a simple mathematical model [13] that best fits the structural (axon number) and functional (PERG amplitude) time courses. In the example of Figure 3A the parameters that best fit the curves are decay rate **b** = 0.3/month, dysfunction coefficient **d** = 0.5 of normal, and sick-to-dead time **τ** = 6.5 months. Using these parameters, it is possible to estimate at each timepoint the proportion of healthy, sick, and dead RGCs (Figure 3B). Although the simple model shown in Figure 3B has obvious limitations, it is useful to show that by the time RGCs start dying at about 7.5 months of age, most RGCs are sick and there are fewer healthy RGCs left. By 10 months of age there are no heathy RGGs left, while there are fewer sick RGCs to repair together with a growing population of dead RGCs. This has implications for choosing the appropriate time window for preventing RGC dysfunction (Rx_t1 in Figure 1), preventing RGC dysfunction and repairing ongoing RGC dysfunction (Rx_t2 in Figure 1), or limiting the rate of RGC death (Rx_t3, Rx_t4 in Figure 1). Neuroprotective strategies in different time windows do not necessarily use similar pharmacological approaches and may result in distinctive outcomes for residual RGC function and RGC number. Analogue models to that shown in Figure 3 may be hypothesized for a variety of conditions impacting the susceptibly and lifespan of RGCs together with their ability to generate electrical signals under a protracted degenerative process. Longitudinal clinical data in early glaucoma patients [27] also show progressive loss of PERG signal, anticipating comparable loss of retinal nerve fiber thickness by several years. The rate of progressive PERG loss in glaucoma suspects may be reduced with IOP-lowering treatment [38]. In human LHON, sudden and severe visual loss often begins with one eye first, usually followed by similar loss in the fellow eye few months later [16]. In unilateral LHON cases, the PERG signal is much altered not only in the symptomatic eye, but also in the asymptomatic eye [39]. This suggests that in the asymptomatic eye there is manifest RGC dysfunction preceding RGC death that may be potentially prevented with a timely neuroprotective intervention, including gene therapy [40,41]. It is conceivable that PERG testing in LHON carriers may anticipate conversion from asymptomatic to symptomatic stage and thus inform timing of neuroprotective therapy.

## 5. Saving RGC Function vs. Saving RGC Bodies

Neuroprotection refers to the relative preservation of neuronal structure and/or function independently of the primary cause of neuronal insult [9]. Ideally, neuroprotection should extend the lifespan of functional RGCs, but this may not always be the case. In principle, neuroprotective strategies that target downstream molecular pathways of cell death such as caspases [42] may keep RGCs on life support for a long time, but these RGCs are not expected to be fully functional. In contrast, strategies that enhance RGC function in the short term do not necessarily alter the rate of progression and may even accelerate RGC death [43]. Noninvasive electrophysiology such as PERG provides the necessary functional outcome to assess the ability of RGCs to generate electrical signals under a protracted degenerative process with or without the presence of neuroprotective treatments. Notably, the PERG can provide a unique contribution to document altered dynamics of RGC function before the tipping point (critical period in Figure 1), which would also represent a rationale for early treatment.

### 5.1. RGC Excitability

The PERG signal depends not only on the presence of functional RGCs, but also on the molecular environment that controls neuronal excitability, such as neurotrophic factors [44]. For example, BDNF/TrkB interaction controls RGC intrinsic excitability by shifting polarization of the membrane potential [45,46]. In healthy mice, a retrobulbar injection of lidocaine (axon transport blocker) does not induce RGC death but rapidly and reversibly reduces the PERG signal [47] (Figure 4). These effects are believed to be induced by deficiency of retrograde signaling in the optic nerve, in particular shortage of neurotrophic factors derived from brain targets via retrograde axonal transport [48]. Axon transport defects are known to play a critical role in the early stage in neurodegenerative disease [49] including glaucoma and LHON [50,51]. Early PERG impairment in glaucoma and optic neuropathies may be at least in part due to altered axonal transport that reduces RGC excitability.

Changes in RGC excitability are reflected in the dynamics of the PERG response [30]. In the normal mouse, the PERG amplitude increases with increasing contrast approximately in a linear manner (i.e., the PERG amplitude at 20% contrast is about 20% of the amplitude at 100% contrast). Although there are measurable differences in PERG contrast dependence in different mouse strains [52] a strong departure from linearity occurs when availability of neurotrophic factors is altered [30]. Figure 5 shows that in naive C57BL/6J mice, the PERG amplitude at 20% contrast is much lower than that at 100% contrast. In C57BL/6J mice receiving an intravitreal injection of BDNF or in C57BL/6J mice who had a chronic lesion of the superior colliculus—resulting in a compensatory upregulation of endogenous BDNF in the retina [53]—the PERG amplitude at low contrast is higher than that of control C57BL/6J mice. Although the mechanisms underlying altered PERG contrast dependence (neurotrophic support/expression, synaptic transmission, plasticity) are only conjectural, changes of PERG dynamics can be used to detect and monitor early changes in RGC excitability.

### 5.2. RGC Adaptation

Rapid dilation of retinal vessels in response to flickering light or fast-reversing patterns (functional hyperemia) is a well-known autoregulatory response driven by increased neural activity in the inner retina [54]. Sustained metabolic stress may in turn influence RGC function, and this is reflected in a progressive decline of the PERG signal to a plateau (adaptation) over 2–4 min [55] (Figure 6). PERG adaptation occurs in mice [33] as well as in humans [56], and represents an index of normal neurovascular autoregulation triggered by a metabolic challenge. PERG adaptation may be reduced or absent when RGCs are dysfunctional as in glaucoma [57] or in optic neuritis [58]. For hypothesis-testing purposes, several models can explain the PERG adaptation dynamics. The model sketched in Figure 6C is based on an energy budget model in a neurovascular-glial network that can be reduced to a simple electrical circuit and mathematical equation [14].

Independently of the underlying mechanisms, PERG adaptation dynamics can be used to detect and monitor altered autoregulation of RGCs together with the neurovascular-glial network impinging on them. As shown in Figure 3, in DBA/2J glaucoma the PERG amplitude progressively decreases with increasing age followed by loss of RGCs [28,29]. In DBA/2J mice, retinal levels of nicotinamide adenine dinucleotide (NAD+, a key molecule in energy and redox metabolism) decrease with age and render aging neurons vulnerable to disease-related insults [6]. Oral administration of the NAD+ precursor nicotinamide (vitamin B3) spares RGCs and their function at older ages [6]. The magnitude of PERG adaptation also decreases with increasing age (Figure 7) [59]. However, prophylaxis with a diet rich in vitamin B3, in addition to saving functional RGCs, also spares the PERG autoregulatory dynamic range in response to flicker [59].

### 5.3. RGC Susceptibility to Stress

Stress tests such as physical exercise are widely employed to investigate altered heart dynamics and are also used in eye diseases. Recovery of vision and VEP amplitude after exposure to a bright light (photostress) may be prolonged in macular diseases [60] and in optic neuritis [61]. Temporary IOP elevation can be induced with head-down (HD) body posture. In DBA/2J mice of different ages, head-down (HD) tilt of 60 degrees causes an IOP elevation of about 5 mm Hg [31]. The PERG of young mice is unaffected by HD, but it becomes substantially depressed in older mice even before the onset of RGC death, suggesting susceptibility to HD stress [31]. In human subjects, HD tilt of 10 degrees induces IOP elevation of about 3 mm Hg on average [32]. While the PERG of normal subjects is not altered by HD tilt, it becomes substantially depressed in a subpopulation of glaucoma suspects [32]. Longitudinal observation of HD-susceptible glaucoma suspects has shown that most of them developed RNFL thinning over 5 years [34].

## 6. Conclusions

Noninvasive, longitudinal assessment of RGC function appears to be a needed diagnostic tool in optic neuropathies. A substantial body of evidence supports the use of PERG to assess the ability of RGCs to generate electrical signals under a protracted degenerative process with or without the presence of neuroprotective treatments, which may have both diagnostic and prognostic values. Further, the PERG can provide a unique contribution to document altered dynamics of RGC function in response to stimuli of different intensity and under different physiological stressors, which may occur before the tipping point and provide the rationale for early treatment. Indeed, a goal of neuroprotective approaches should be preserving and restoring RGC integrity. The PERG can also be useful to screen acute neuroenhancement and toxic effects of neuroprotective drugs. User-friendly versions of the PERG technology are now commercially available for both clinical and experimental use.

## Figures and Tables

**Figure 1 ijms-23-05751-f001:**
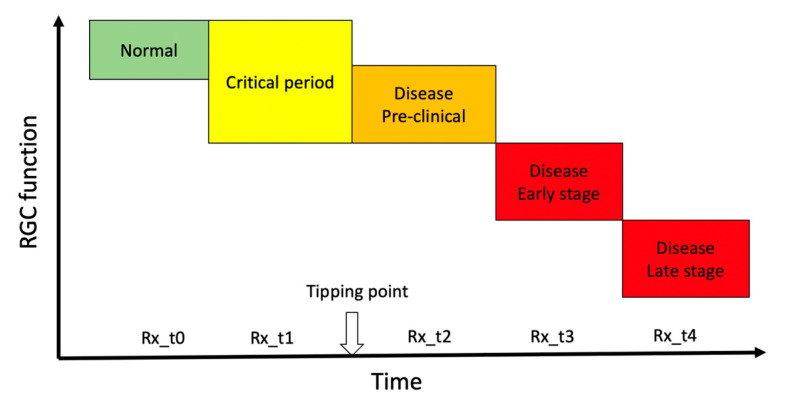
Hypothetical transitional stages of retinal ganglion cell function over the course of a progressive optic neuropathy. The vertical arrow on the time axis represents the idealized transition from a physiological state to a pathological state (tipping point). Abbreviations on the time axis (Rx_t0-t4) represent the idealized time windows of neuroprotective treatment over the course of the disease. Rx_t0 represents prophylaxis. In the time window immediately preceding the tipping point (critical period), early pathological processes may induce RGC dysfunction that is identifiable using sensitive electrophysiological tests such as PERG. Neuroprotective treatment during the critical period (Rx_t1) may interrupt the pathological process and even reverse RGC dysfunction. Neuroprotective treatments at increasing time after the tipping point (Rx_t2, Rx_t3, Rx_t4) may slow progression of RGC dysfunction.

**Figure 2 ijms-23-05751-f002:**
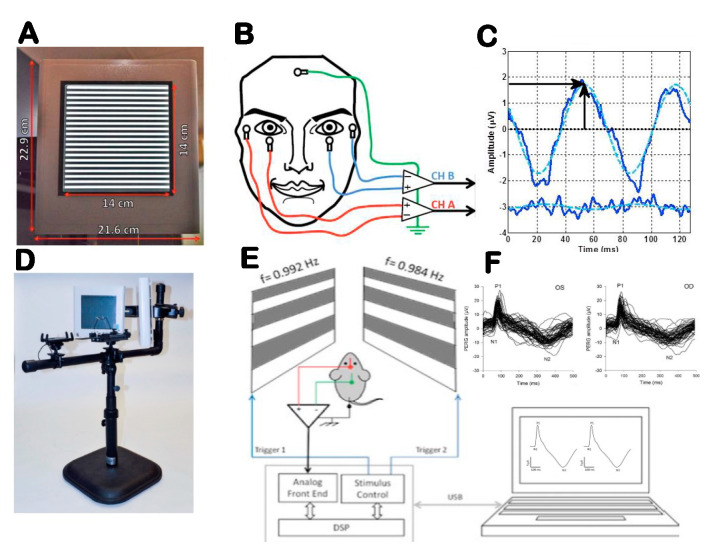
Outline of PERG setup for clinical (**A**–**C**) and experimental (**D**–**F**) use. (**A**) Pattern stimulus. (**B**) Taped skin electrode placement. (**C**) Example of steady-state PERG and noise waveforms (continuous lines) in response to one stimulus cycle (two pattern reversals). The dashed sinusoidal waveforms superimposed to the SS-PERG waveforms represent the frequency-domain component that is measured in amplitude (vertical arrow) and latency (horizontal arrow). (**D**) Tiltable pattern displays to change body posture. (**E**) Binocular pattern stimuli reverse at slightly different frequency to retrieve uniocular PERG using a common subcutaneous needle in the snout. (**F**) Examples of PERG waveforms recorded simultaneously for both eyes consisting of a major positive component (P1) at 80–100 ms and a negative component (N2) at about 350 ms. Amplitude is measured from P1 to N2, and latency is measured as time-to-peak of P1. From Monsalve et al., 2017 and Chou et al., 2014.

**Figure 3 ijms-23-05751-f003:**
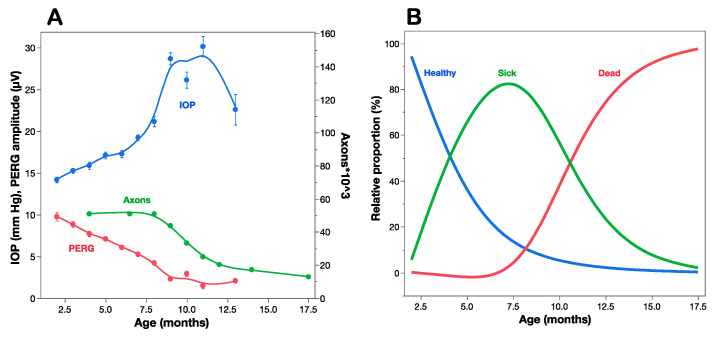
Structure–function relationships in DBA/2J mice. (**A**) IOP, PERG amplitude, and optic nerve axon counts as a function of age of mice. (**B**) Estimated proportion of healthy, sick, and dead retinal ganglion cells at different ages based on a mathematical model that accounts for the structural/functional differences shown in panel (**A**). (**A**) replotted from Saleh et al., IOVS 2007; (**B**) replotted from Porciatti and Chou, Cells 2021.

**Figure 4 ijms-23-05751-f004:**
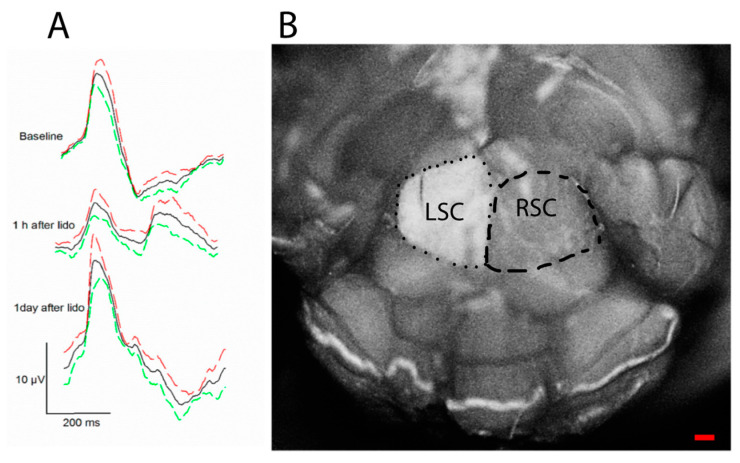
Retrobulbar lidocaine blocks axon transport and reversibly reduces PERG amplitude. The axon transport marker Alexa Fluor 488 Cholera Toxin B was intravitreally injected in both eyes of young DBA/2J mice, and lidocaine was immediately injected in the retrobulbar space of the left eye. (**A**) The PERG signal (average of 5 mice ± SE) was much reduced 1 h after retrobulbar lidocaine compared to baseline and fully recovered one day after. (**B**) After PERG recording, the entire brain was fixed, and both superior colliculi (SC) exposed by aspirating the overlying cortex. The dorsal view shows the surface of both left SC (LSC) and right SC (RSC), roughly outlined with a dotted line and a dashed line, respectively. Confocal scanning laser ophthalmoscopy was performed to identify Alexa Fluor 488 Cholera Toxin B fluorescence. Note the bright fluorescence of the LSC and the absent fluorescence of the RSC, indicating blockage of axon transport along the retinocollicular pathway of the lidocaine-treated eye; scale bar, 200 µm. Replotted from Chou et al., Int. J. Mol. Sci. 2018.

**Figure 5 ijms-23-05751-f005:**
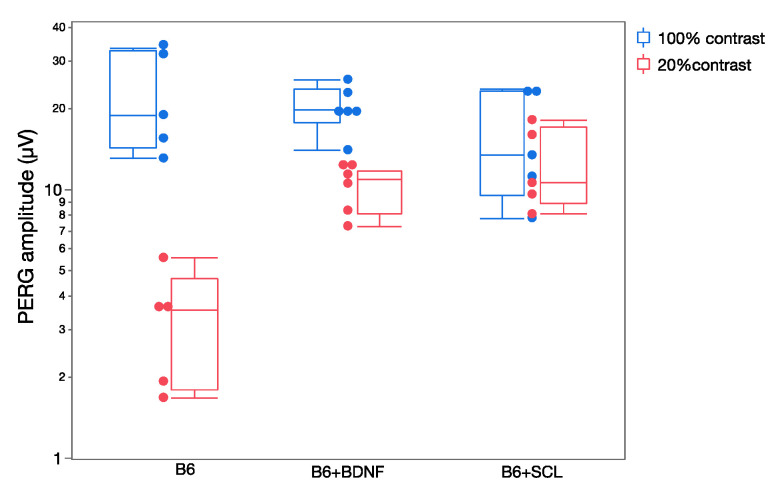
Contrast dependence of PERG amplitude in control C57BL/6J (B6) mice and in B6 mice in which neurotrophic support has been altered ((B6 + BDNF): intravitreal injection of BDNF; (B6 + SCL): chronic lesion of the contralateral superior colliculus). Replotted from Chou et al., Exp. Eye Res. 2016.

**Figure 6 ijms-23-05751-f006:**
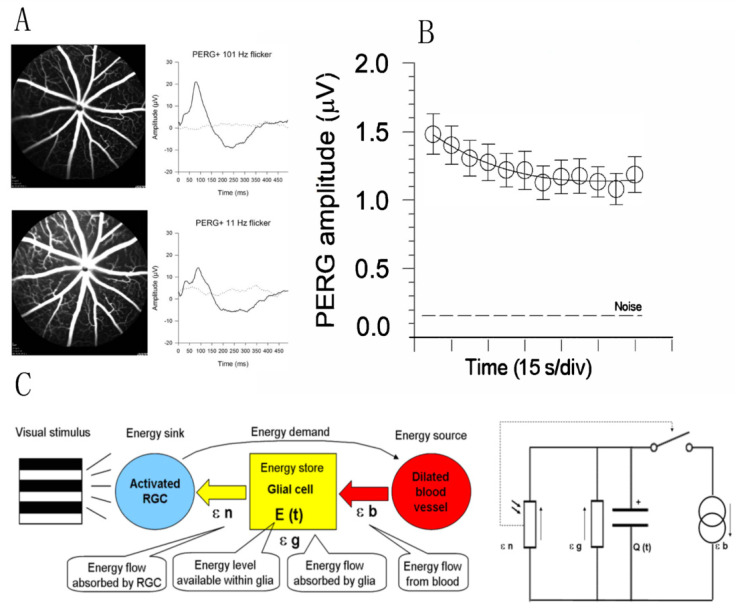
Flicker-induced PERG adaptation. (**A**) Flickering light at 11 Hz superimposed to a pattern stimulus induces vasodilation in C57BL/6J mouse, as shown by fluorescein angiography, and reduces the PERG amplitude compared to the same pattern stimulus with superimposed flicker at 101 Hz (invisible to photoreceptors). (**B**) In human subjects, the PERG signal in response to sustained pattern reversal at 16 rps becomes progressively reduced to a plateau over 2 min. (**C**) Energy budget model that accounts for the temporal dynamics of PERG adaptation in mice and human subjects. At any given time, the energy available to activated neurons (ε n, photoresistor) depends on the energy flow provided by glial stores (ε g capacitor) and vascular supply (ε b, current generator) minus the energy absorbed in the process (ε g, resistor). The switch connecting activated neurons to vascular supply represents the neurovascular coupling. The direction of arrows indicates the energy flow. (**A**) Replotted from Chou et al., Sci. Rep. 2019; (**B**) replotted from Porciatti et al., IOVS 2005; (**C**) replotted from Porciatti and Ventura, Vis. Res. 2009.

**Figure 7 ijms-23-05751-f007:**
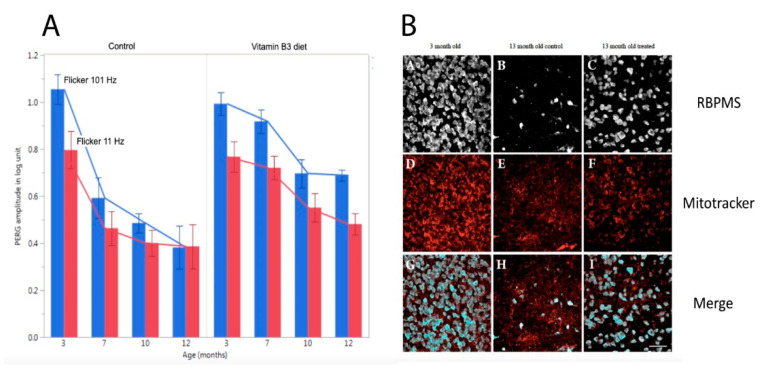
Flicker-PERG amplitude adaptation and RGC survival in control DBA/2J mice fed with standard diet and in DBA/2J mice fed with vitamin B3-enriched diet. (**A**) Amplitude of PERG recorded with superimposed flickering light at either 101 Hz (baseline, blue bars) or 11 Hz (test, red bars) in mice of different ages. The difference between baseline and test represents the magnitude of flicker-PERG adaptation. Note in control mice that flicker-PERG adaptation progressively decreases with increasing age, whereas in vitamin B3-treated mice PERG adaptation is preserved together with a slower decline of PERG amplitude with age. Error bars represent the SEM (*n* = 10 for each group). (**B**) Representative examples of RBPMS (RNA-binding protein with multiple splicing)-immunostained RGCs and Mitotracker-immunostained mitochondria in flat mounted retinas of DBA/2J mice 3 months old and 13 months old fed with either standard diet or vitamin B3-enriched diet. Note the rescue effect of vitamin B3-enriched diet on RGCs and mitochondria in mice 13 months old. Replotted from Chou et al., Nutrients 2020.

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
