# Peer review of "Using Noninvasive Electrophysiology to Determine Time Windows of Neuroprotection in Optic Neuropathies"

_ijms, 2022, doi:10.3390/ijms23105751_

Round 1

Reviewer 1 Report

Porciatti and Chou wrote a review on the time windows of neuroprotection in optic neuropathies. They suggested using a non-invasive electrophysiological technique termed pattern electroretinogram (PERG) to assess the ability of retinal ganglion cells to generate electrical signals. The review is interesting and the technique can be used in both in vivo and clinical settings.

Some major concerns need to be addressed to allow for this specialized topic to be suitable for a general audience of int. J. Mol. Sci. journal:

(order of appearance in the review)

  • The title is too vague and needs to focus more on the main story of the review. For example, suggest ‘Using non-invasive techniques to determine time windows of neuroprotection in optic neuropathies’.
  • (Line 45) Need to provide quantitative data on how long and short the diseases are.
  • (lines 75-76) This sentence suggests neuroprotective treatment worsens the outcome, which is incorrect as the technique is only for diagnostic purposes.
  • (Section 3) Require to explain how PERG works and its effectiveness as the technique measures the skin around the face to measure the RGCs. What does PERG measure exactly? Also, need examples to support this technique. This is important as the whole review is based on this technique to obtain the time window for neuroprotection.
  • (Figure 3) With question 4 above. Figure 3A suggests the PERG detection is via increased IOP. What happens if the damage is not related to a raised IOP? If it has no effect, then can the review be as general as a review on optic neuropathies? Furthermore, when do the symptoms become visible? This is important as it indicates how much neuronal loss occurs before symptoms occur.
  • (lines 157-159). If it is known that losing one eye in LHON, the other eye's vision will be lost after a few more months, then what is the need to determine if the other asymptomatic eye will suffer from RGC dysfunction?
  • (figure 4). Need arrow and arrowhead to indicate regions in B. Also, need a schematic/diagram to show where B is obtained from the whole brain.
  • (Page 6) Avoid using the B6 abbreviation for C57Bl/6J as unconventional and make understanding difficult.
  • It would be useful to briefly explain why vitamin B3 was chosen for study.
  • (Figure 6) What was the bahavioural response as the RGC may be present but they could be non-functional. Should delay treatment to see when it doesn’t work as this would be more clinically relevant. What is the staining used for panel B and unsure what ‘!”#$%’ means.
  • (Conclusion) The first sentence suggests PERG is involved in the neuroprotection but it is more a diagnostic technique, so not directly involved.

Minor concerns:

  • (line 79) Define ERG
  • Grammar/spelling (line 105) an, (line 159) Fellow, (line 160) non
  • (line 208) Unsure what is 30?

Author Response

Porciatti and Chou wrote a review on the time windows of neuroprotection in optic neuropathies. They suggested using a non-invasive electrophysiological technique termed pattern electroretinogram (PERG) to assess the ability of retinal ganglion cells to generate electrical signals. The review is interesting and the technique can be used in both in vivo and clinical settings.

—thank you for the words of appreciation

Some major concerns need to be addressed to allow for this specialized topic to be suitable for a general audience of int. J. Mol. Sci. journal:

— We appreciate the thorough review and constructive criticisms

(order of appearance in the review)

The title is too vague and needs to focus more on the main story of the review. For example, suggest ‘Using non-invasive techniques to determine time windows of neuroprotection in optic neuropathies’.

— Good suggestion. We have changed the title as ' Using non-invasive electrophysiology to determine time windows of neuroprotection in optic neuropathies'

(Line 45) Need to provide quantitative data on how long and short the diseases are.

— We have now specified the approximate duration of the critical period in glaucoma and LHON

(lines 75-76) This sentence suggests neuroprotective treatment worsens the outcome, which is incorrect as the technique is only for diagnostic purposes.

—We agree and have now rephrased the legend of Figure 1 to clarify it.

(Section 3) Require to explain how PERG works and its effectiveness as the technique measures the skin around the face to measure the RGCs. What does PERG measure exactly? Also, need examples to support this technique. This is important as the whole review is based on this technique to obtain the time window for neuroprotection.

—Good suggestion. We have now expanded Section 3 to include a short paragraph on the PERG generators. Please consider that this matter is also addressed in Section 5.

(Figure 3) With question 4 above. Figure 3A suggests the PERG detection is via increased IOP. What happens if the damage is not related to a raised IOP? If it has no effect, then can the review be as general as a review on optic neuropathies? Furthermore, when do the symptoms become visible? This is important as it indicates how much neuronal loss occurs before symptoms occur.

—Great question. The DBA/2J model is a chronic model that is useful to establish structural-functional relationships over time. It is true that IOP plays a significant role in progressive PERG loss, although it is relatively small compared to age. Multiple regression analysis of data shown in Figure 3A compares significance of IOP (Log P= 3.8) with age (Log P=28.5). This is now specified in the text. We described another chronic mouse models showing early and progressive PERG loss with partially degenerated axons in old transgenic mice without IOP elevation (IOVS 2014, PMID: 25125600: Transgenic Mice Expressing Mutated Tyr437His Human Myocilin Develop Progressive Loss of Retinal Ganglion Cell Electrical Responsiveness and Axonopathy With Normal IOP). However, there are no sufficient data at intermediate time points in this transgenic mouse for modeling a structural-functional relationship.

The question of the temporal relationship between PERG loss and symptoms is a very important one. However, we do not think that this can be suitably addressed in mouse models. In human patients there are numerous sources of evidence that the PERG can be altered in glaucoma suspects before any measurable loss in RNFL thickness or visual field sensitivity. In unilateral LHON patients, the PERG can be much altered in the asymptomatic eye (PMID: 35344016).

(lines 157-159). If it is known that losing one eye in LHON, the other eye's vision will be lost after a few more months, then what is the need to determine if the other asymptomatic eye will suffer from RGC dysfunction?

—Good point. It is true that in unilateral LHON, impending vision loss in the asymptomatic eye is predictable. However, the recent finding that the PERG is much altered in the asymptomatic eye of LHON patients (PMID: 35344016) suggests that altered PERG is a reliable predictor of visual loss in LHON and can be used to monitor LHON carriers to establish the necessity of neuroprotective treatment in the hope to prevent vision loss.

(figure 4). Need arrow and arrowhead to indicate regions in B. Also, need a schematic/diagram to show where B is obtained from the whole brain.

— The figure and legend have been updated to explain how the brain tissue was obtained and to delineate the superior colliculi.

(Page 6) Avoid using the B6 abbreviation for C57Bl/6J as unconventional and make understanding difficult.

— The abbreviation for C57BL/6J has been eliminated in the text but it has been maintained in figure 5 to simplify the legend on the x-axis

It would be useful to briefly explain why vitamin B3 was chosen for study.

— We introduced a statement in the text to clarify this point. Thank you for the suggestion

(Figure 6) What was the bahavioural response as the RGC may be present but they could be non-functional. Should delay treatment to see when it doesn’t work as this would be more clinically relevant. What is the staining used for panel B and unsure what ‘!”#$%’ means.

— This is an interesting question. As shown in the model of Figure 3 and discussed in Section 5, sick RGCs still function although with a reduced electrical responsiveness, and this is reflected in an altered PERG signal. Sick RGCs may still sustain a behavioral response. While this is hard to demonstrate behaviorally in mouse models, it is demonstrated in reports of PERG abnormality but normal visual acuity and visual field in subjects with pre-perimetric glaucoma and asymptomatic eyes of unilateral LHON patients. We believe that the clinical interest of the PERG relies on detection of reduced ERG responsiveness in pre-symptomatic eyes that has predictive value and may orient therapeutic management of patients.

RBPMS (RNA-binding protein with multiple splicing) is a protein that specifically stain all RGCs. This is now included in the legend.

(Conclusion) The first sentence suggests PERG is involved in the neuroprotection but it is more a diagnostic technique, so not directly involved.

— We agree. The statement has been rephrased.

Minor concerns:

(line 79) Define ERG

—done

Grammar/spelling (line 105) an, (line 159) Fellow, (line 160) non

—done

(line 208) Unsure what is 30?

— we added brackets missing in reference #30

Reviewer 2 Report

Authors offer a perspective on how non-invasive, longitudinal assessment of RGC function appears to be a necessary component of neuroprotective strategies in optic neuropathies. They discuss how PERG can assess the ability of RGCs to generate electrical signals under a protracted degenerative process, which may have both diagnostic and prognostic values and provide the rationale for early treatment.

Although this Perspective could be of interest for the journal an important modification has to be introduced before being accepted for publication:

  • Lines 135-149. This part should be rewritten and explained in a more understandable way. Authors mention a mathematical model and introduce several parameters associated to figure 3B that are difficult to understand if you are not familiar with the model.

Author Response

Authors offer a perspective on how non-invasive, longitudinal assessment of RGC function appears to be a necessary component of neuroprotective strategies in optic neuropathies. They discuss how PERG can assess the ability of RGCs to generate electrical signals under a protracted degenerative process, which may have both diagnostic and prognostic values and provide the rationale for early treatment.

 Although this Perspective could be of interest for the journal an important modification has to be introduced before being accepted for publication:

—Thank you for the thorough review and for pointing out the strengths and weaknesses of our manuscript. We have accepted the constructive criticism and have edited the manuscript accordingly.

Lines 135-149. This part should be rewritten and explained in a more understandable way. Authors mention a mathematical model and introduce several parameters associated to figure 3B that are difficult to understand if you are not familiar with the model.

—We agree and edited the text to include the conceptual principles on which the model is based and to clarify the model.

Round 2

Reviewer 1 Report

All concerns have been addressed accordingly.